# Mining the Proteome of *Toxoplasma* Parasites Seeking Vaccine and Diagnostic Candidates

**DOI:** 10.3390/ani12091098

**Published:** 2022-04-23

**Authors:** Sajad Rashidi, Javier Sánchez-Montejo, Reza Mansouri, Mohammad Ali-Hassanzadeh, Amir Savardashtaki, Mohammad Saleh Bahreini, Mohammadreza Karimazar, Raúl Manzano-Román, Paul Nguewa

**Affiliations:** 1Department of Parasitology and Mycology, School of Medicine, Shiraz University of Medical Sciences, Shiraz 7134845794, Iran; sajaderashidiii@gmail.com (S.R.); bahreinimohammadsaleh@gmail.com (M.S.B.); mkarimazar91@gmail.com (M.K.); 2Infectious and Tropical Diseases Group (e-INTRO), Institute of Biomedical Research of Salamanca-Research Center for Tropical Diseases at the University of Salamanca (IBSAL-CIETUS), Faculty of Pharmacy, University of Salamanca, 37008 Salamanca, Spain; s.montejo@usal.es; 3Department of Immunology, Faculty of Medicine, Shahid Sadoughi University of Medical Sciences and Health Services, Yazd 8915173143, Iran; rmansouri@ssu.ac.ir; 4Department of Immunology, School of Medicine, Jiroft University of Medical Sciences, Jiroft 7861615765, Iran; m.hassanzadeh@jmu.ac.ir; 5Department of Medical Biotechnology, School of Advanced Medical Sciences and Technologies, Shiraz University of Medical Sciences, Shiraz 7134845794, Iran; dashtaki63@gmail.com; 6Department of Microbiology and Parasitology, ISTUN Institute of Tropical Health, IdiSNA (Navarra Institute for Health Research), University of Navarra, c/Irunlarrea 1, 31008 Pamplona, Spain

**Keywords:** *Toxoplasma gondii*, toxoplasmosis, targets, vaccines, diagnostics

## Abstract

**Simple Summary:**

The One Health concept to toxoplasmosis highlights that the health of humans is closely related to the health of animals and our common environment. Toxoplasmosis outcomes might be severe and fatal in patients with immunodeficiency, diabetes, and pregnant women and infants. Consequently, the development of effective vaccine and diagnostic strategies is urgent for the elimination of this disease. Proteomics analysis has allowed the identification of key proteins that can be utilized in the development of novel disease diagnostics and vaccines. This work presents relevant proteins found in the proteome of the life cycle-specific stages of *Toxoplasma* parasites. In fact, it brings together the main functionality key proteins from *Toxoplasma* parasites coming from proteomic approaches that are most likely to be useful in improving the disease management, and critically proposes innovative directions to finally develop promising vaccines and diagnostics tools.

**Abstract:**

*Toxoplasma gondii* is a pathogenic protozoan parasite that infects the nucleated cells of warm-blooded hosts leading to an infectious zoonotic disease known as toxoplasmosis. The infection outcomes might be severe and fatal in patients with immunodeficiency, diabetes, and pregnant women and infants. The One Health approach to toxoplasmosis highlights that the health of humans is closely related to the health of animals and our common environment. The presence of drug resistance and side effects, the further improvement of sensitivity and specificity of serodiagnostic tools and the potentiality of vaccine candidates to induce the host immune response are considered as justifiable reasons for the identification of novel targets for the better management of toxoplasmosis. Thus, the identification of new critical proteins in the proteome of *Toxoplasma* parasites can also be helpful in designing and test more effective drugs, vaccines, and diagnostic tools. Accordingly, in this study we present important proteins found in the proteome of the life cycle-specific stages of *Toxoplasma* parasites that are potential diagnostic or vaccine candidates. The current study might help to understand the complexity of these parasites and provide a possible source of strategies and biomolecules that can be further evaluated in the pathobiology of *Toxoplasma* parasites and for diagnostics and vaccine trials against this disease.

## 1. Introduction: Vaccine and Diagnostic Strategies in Toxoplasmosis

*Toxoplasma gondii*, the causative agent of toxoplasmosis, is a pathogenic protozoan parasite that infects the nucleated cells of warm-blooded hosts [1]. Toxoplasmosis affects approximately one-third of the world’s human population but also may be a concern in a considerable number of mammalian and avian species, with potential associated public health risks [2,3,4]. Although toxoplasmosis is usually asymptomatic in immune-competent individuals, the outcomes of infection could be severe or fatal in patients with immunodeficiency, diabetes patients, and pregnant women and infants [5,6,7]. The One Health approach to toxoplasmosis highlights that the health of humans is closely related to the health of animals and our common environment. Therefore, the development of effective vaccine and diagnostics strategies is urgent for the elimination of this infection.

The immunological effects of numerous vaccination trials, including attenuated and inactivated vaccines, genetically engineered vaccines, subunit vaccines, and DNA vaccines, have been evaluated and developed against toxoplasmosis in animal models. However, such strategies have also encountered several difficulties, such as vaccine construct, routes of administration, and standardization of immunization evaluation [8].

Live attenuated vaccines are more likely to produce the beneficial T helper (Th1) immune response compared to the subunit or DNA vaccines in different infectious agents, especially in intracellular pathogens. However, there are a limited number of trials evaluating concerns of such vaccines, maybe due to their reversion of attenuated pathogens to their virulent form [9,10]. Therefore, whole genome sequencing and appealing strategies, including clustered regularly interspaced short palindromic repeats (CRISPR)-Cas9, to edit genes of *Toxoplasma* parasites and the construction of novel mutant strains have recently accelerated the improvement of live attenuated vaccines against toxoplasmosis. Thus, a huge range of experimental live-attenuated vaccines were described through deleting or knocking-out numerous genes [11,12]. Recently, a wide range of DNA vaccines against toxoplasmosis have been developed. The efficacy of such vaccines is deeply affected by the method of vaccine delivery. Furthermore, it has been suggested that DNA vaccines expressing several antigens could further induce protection against toxoplasmosis than single antigen vaccines [11]. Similarly, numerous candidate proteins involved in the *Toxoplasma* parasite pathogenesis, survival, and relevant critical pathways to this infection, have been employed as vaccine antigens in recombinant subunit vaccines. In this sense, the protective efficacy of multi-antigenic subunit vaccines has been underlined [11].

Some of the *Toxoplasma* proteins, such as calcium-dependent protein kinases (CDPKs), have been targeted as vaccine candidates against toxoplasmosis by using the abovementioned strategies. It has been indicated that the use of CDPK1, CDPK2 and CDPK3 in the form of attenuated, recombinant and DNA vaccine induced high levels of Th1-associated cytokines, and prolonged survival and decrease brain cysts in vaccinated mice [13,14,15,16,17]. In addition, in silico tools have predicted recently potential immunogenic B- and T-cell epitopes for CDPK4 and CDPK7, pointing out their potential as appropriate vaccine candidates against *T. gondii* [18,19].

Despite of all those approaches, there is currently no licensed toxoplasmosis vaccine available for humans. “Toxovax” is considered as the only commercial vaccine (designed based on live attenuated (S48 strain) strategy) against congenital toxoplasmosis in ewes. The use of such vaccines for humans needs to overcome big challenges, such as their high cost, adverse effects, short shelf-life, and the risk of reverting to a virulent form.

On the other hand, different serodiagnostic tests have been expanded for the detection of human toxoplasmosis. As a recent strategy, the use of recombinant proteins or a combination of several recombinant proteins have been successfully suggested for measuring *T. gondii* antibodies at different stages of this infection. However, the development of relatively rapid, highly sensitive and specific methods has remained a prominent challenge in this sense. Therefore, integrating genomic, transcriptomic, and proteomic tools and multilocus genotyping methods with molecular and bioinformatics techniques have been currently suggested to increase the sensitivity and specificity of the diagnostic methods based on the use of recombinant proteins [20].

## 2. *Toxoplasma* Life Cycle Stages Proteome and Proteomics

The different *T. gondii* life stages employ specific mechanisms for triggering stage conversion, and these could be related to pathobiology within the host [21]. Changes in the expression levels of some proteins also occur as parasites progress through their life cycles, and it is likely that particular proteins have important functions in restricted life stages [22]. Since most of such proteins are involved in the parasite survival, virulence and modulation of the host immune response, the identification and biological understanding of these critical proteins in the different parasitic forms might be useful for the diagnosis, directed targeting and prevention of the disease (vaccination). Accordingly, further improvements of the effective serodiagnostic tools and the potentiality of vaccine candidates in inducing the host immune responses are considered key factors for the discovery of new functionality relevant proteins in the *Toxoplasma* (life cycle stages) proteome [3].

Some of the differentially expressed proteins in each stage of the *Toxoplasma* parasite’s life cycle could be correlated with the pathogenesis or might induce host immune responses. Thus, the use of such a common immunodominant protein expressed in all stages or the selection of several immunodominant proteins in all stages as a multistage (multivalent) vaccine could efficiently induce the desired immune responses during all stages of the parasite. On the other hand, designed vaccines based on a multistage strategy are able to exhibit efficient effects in initial and recurrent infections and probably exert major functions in restricting the bradyzoites released from tissue cysts [23]. Due to the active functions of proteins, such as *Toxoplasma* dense granule antigen 1 (TgGRA1) and bradyzoite antigen 1 (BAG1), during the invasion of host cell and their potential to induce increased immunoglobulin G (IgG) levels (with slight tendency to IgG2a response) and interferon gamma (IFN-γ) secreting cluster of differentiation 4 (CD4) and CD8 cells, these proteins were described as vaccine candidates to generate a multistage vaccine which could block the tachyzoite and bradyzoite stages of the parasite [24]. A common immunodominant protein, such as microneme protein 3 (MIC3), that could be expressed as a critical protein in the proteome of *Toxoplasma*-oocysts [22], *Toxoplasma*-tachyzoites (excretory-secretory antigens (ESAs) and soluble tachyzoite antigens (STAgs)) [25,26] might also be a potential multistage vaccine against toxoplasmosis. Evidence has shown that MIC3, 4, 13, rhoptry neck protein 5 (RON5), rhoptry protein 2 (ROP2), and GRA1, 6, 8, 14 with potential pathogenicity and immunogenicity properties were expressed in the three infective stages of *Toxoplasma* parasites. Moreover, other potential virulent and immunodominant proteins, including rhomboid 4 (ROM4), ROP5, 16, 17, 38, GRA2, 4, 15, 10, 12, 16, RON4, MIC1, 5, and surface antigen 3 (SAG3), were identified only in tachyzoites and bradyzoites stages [23]. These proteins could also be considered in multistage vaccine against toxoplasmosis.

Proteomics analysis has allowed the identification of key proteins that can be utilized in the development of novel disease diagnostics and vaccines [27,28,29,30,31]. Thus, proteomic approaches may help to identify such proteins with crucial roles in mediating parasite capacity to modulate the host immune response. Those strategies may enable us to detect and select promising vaccine and diagnostic targets against toxoplasmosis [32,33,34]. Therefore, the aim of this study is bringing together the main functionality relevant proteins from *Toxoplasma* parasites coming from proteomic approaches most likely to be useful in improving disease management and to critically propose innovative directions to finally develop promising vaccines and diagnostic tools. Accordingly, this work also covers the possible vaccine and diagnostics properties of such important proteins.

## 3. Vaccine and Diagnostic Proteins Identified in the Proteome of *T. Gondii* Developmental Stages

A number of proteomic techniques have been used for the study of the proteins expressed in the life cycle-specific stages of *T. gondii* (Figure 1) [22,35,36,37,38]. This approach leads to the identification of relevant stage-specific proteins in tachyzoite [37,38,39,40,41,42,43,44], tachyzoite STAgs/ESAs [26,45,46], oocyst [22,47], cyst [48,49], and sporocyst/sporozoite [50,51] of the parasite. Here, we present each of the most important proteins and the associated biological functions (Table 1) to understand their potential for seeking and suggesting the plausible *Toxoplasma*-vaccine and diagnostics candidates (Figure 2 and Table 2). According to Figure 2, most of the discussed studies regarding vaccine and diagnostics candidates have been designed based on the recombinant proteins and molecular vaccine (mostly DNA vaccine) strategies [52,53].

### 3.1. Actin Depolymerizing Factor (ADF)

The intranasal immunization with recombinant TgADF (rTgADF) can simultaneously trigger mucosal and systemic immune responses and protect mice against *T. gondii* infection [74]. ADF increased survival rate (36.36%) and decreased tachyzoite burden in the liver (67.77%) and brain (51.01%) from vaccinated mice. The immunostimulatory properties obtained concerning ADF, as an overexpressed protein in type III strain (CTG) [37], might suggest the use of non-pathogenic or attenuated form of *Toxoplasma* parasite as promising for vaccines against toxoplasmosis.

### 3.2. Nucleoside-Triphosphatases (NTPases)

The rTgNTPase-II protein is able to provide protective Th1 cell-mediated immunity against *T. gondii*. The immunogenic potential of a self-amplifying RNA vaccine-encoding TgNTPase-II gene, RREP-NTPase-II, delivered by a synthetic lipid nanoparticle (LNP) has been recently evaluated in a mouse model [75]. Mice vaccinated with RREP-NTPase-II-encapsulated LNP displayed significantly enhanced protection against acute infection as well as chronic infection. The survival time was prolonged and parasite burden in the brain after acute (46.4%) and chronic (62.1%) infections was reduced in vaccinated mice. The results suggest that the combination of self-amplifying RNA and LNP would be beneficial to the development of a safe and long-acting vaccine against toxoplasmosis.

### 3.3. GRAs

DNA vaccination has been performed with genes encoding the proteins GRA1, GRA7, and ROP2; it induced a partial protection against infections caused by different virulent *T. gondii* strains in CH3 mice. A high ratio of specific IG2a (IgG2a) to IgG1 antibodies detected in DNA-vaccinated mice represented a Th1-type response. The survival rate was increased from 50% to at least 90% in most of the vaccinated mice [76].

*Toxoplasma* GRA4 antigen was expressed by chloroplast transformation (chlGRA4) in tobacco plants and examined the cellular and humoral responses and the grade of protection against toxoplasmosis after oral administration in a murine model [77]. The oral immunization with chlGRA4 led to the induction of both a mucosal immune response and a systemic response and a decrease of 59% in the brain cyst load of mice compared to control mice, leading to the control of toxoplasmosis and reduction of parasite load.

Recent experimental data inferred from tachyzoite-GRA5 showed that the recombinant form of this protein can be applied as an antigenic protein for designing serodiagnostic tools to identify toxoplasmosis, especially in hemodialysis patients. The specificity and sensitivity of enzyme-linked immunosorbent assay (ELISA) were 93% and 96%, respectively. The loop-mediated isothermal amplification (LAMP) method also corroborated the accuracy and reliability of the results obtained by designed and commercial ELISA kits [78].

The intramuscular injection of sheep with a DNA liposome formulated plasmid coding for GRA1, GRA4, GRA6 and GRA7 is an effective system that induces a significant immune response against *T. gondii*. GRA7 stimulated a Th1-like immune response, increasing anti-GRA7 IgG2 antibody levels and IFN-γ responses, whereas GRA1, GRA4 and GRA6 induced an IgG1 type antibody response with a limited IFN-γ response [79].

A DNA vaccine based on GRA6 of *T. gondii* can also induce strong humoral and cellular immunity (the major histocompatibility complex (MHC) restricted immune response) and provide partial protection against toxoplasmosis in vaccinated BALB/c mice (increasing serum levels of anti-GRA6 IgG and splenocyte proliferation) [80]. All these data further highlight the appropriate property of GRAs as DNA vaccines for immunity against toxoplasmosis.

It has been recently indicated that the genetic disruption of GRA9 in *Toxoplasma*-type II PLK strains reduced parasite replication, survival, and cyst formation in mice models in vivo. Interestingly, the use of this attenuated vaccine significantly induced full immune responses (inducing high levels of pro-inflammatory cytokine IFN-γ and interleukin-12 (IL-12), maintaining the high *T. gondii*-specific IgG level, and mixed high IgG1/IgG2a levels) and represented 100% protection against acute and chronic *T. gondii* challenges [81].

Moreover, adjuvant and immunogenic potential of an rTg profilin (rTgPF) protein has been recently evaluated in a vaccine formulation in combination with the GRA7 antigen in a murine toxoplasmosis model [82]. The use of this vaccine significantly enhanced immune responses (generating a Th1-biased immunity through the induction of lymphocyte proliferation, the activation of CD4^+^ T cells and an increased IFN-γ production) and protection against chronic toxoplasmosis. TgPF acts as a ligand for toll-like receptor 11 (TLR11) and TLR12, inducing innate immune responses that increase type 1 adaptive responses, therefore highlighting the role of PF as a potential adjuvant in vaccine strategies against toxoplasmosis [82,83]. However, since TgProfilin interacts with TLR receptors that are not present in humans or livestock species [83,84], it appears less useful in this sense for relevant host vaccination.

Recent data revealed that novel and interesting functions for GRA7 and GRA14 in the induction of nuclear factor kappa B (NFκB) (regulating the induction of Th1 immunity) during *Toxoplasma* infection. NFκB activation mediated through GRA7 and GRA14 was correlated with the Th1 response increased by inflammatory cytokines. Consequently, although the parasite survival was increased by changing the active form of parasite to inactive form (tissue cysts), the tissue invasion by parasite was decreased and led to the survival of the host [85]. This information indicated that GRA7 and GRA14 induced host immunity through NFκB and limited parasite expansion and probably further highlights the role of these GRAs in vaccination. The increase in antibody titers (total IgG and IgG2a) and the concentration of IFN-γ (a Th1 type response) was also related to the vaccination by GRA14 adjuvanted with calcium phosphate nanoparticles (CaPNs) [86]. In addition, in silico and bioinformatics approaches also underlined GRA4, GRA7 and GRA14 as possible vaccine candidates against toxoplasmosis [87].

Overall, it seems that among the GRAs, GRA4 and GRA7 could efficiently increase the survival time of vaccinated animals. The combination of GRA3, GRA7 and MIC2-associated protein (M2AP) antigens successfully reduced the cyst burden in vaccinated mice (93.5%). In addition, GRA6 and GRA10 correlated with a high immunogenicity and GRA1 and GRA2 were suggested as important virulence factors and inductors of host immune responses [58].

### 3.4. SAG1

SAG1 has been described as a potential inducer of the host immune system and a vaccine candidate [88]. The nanospheres of rSAG1 were recently found to be a bio-compatible candidate for the development of a vaccine against toxoplasmosis. The intranasal injection of this recombinant protein elevated humoral responses of specific IgA and IgG2a in vaccinated mice [89]. However, it seems that the application of multi-stage antigens or cocktailed vaccines, SAG1 in combination with other proteins, including ROP2, ROP4, GRA1, GRA4, GRA7, MIC3, and BAG1 can be more effective against all stages of the *Toxoplasma* life cycle. ROP2 and SAG2 have been recognized as the most common antigens used for experimental cocktail vaccines together with SAG1 [90]. On the other hand, immunoinformatics-based simulation represented the appropriate interaction of a multi-epitope vaccine construct containing SAG1, along with apicoplast ribosomal proteins (S2, S5 and L11) with human TLR4 and effective induction of humoral (potent stimulation of T- and B-cell mediated immune responses) and, especially, cellular immune responses (developing high levels of IFN-γ and other components of the cellular immune profile) [91].

The use of an ELISA method based on TgrSAG1 was a potential immunodiagnostic tool (sensitivity and specificity of 98.5% and 100%, respectively) that was more accurate and reliable than latex agglutination test (LAT) for the diagnosis of *Toxoplasma* infection in human [92]. Furthermore, the integrated recombinant multi-epitope antigens of *T. gondii* (SAG1, ROP1, and GRA7) might be useful to develop clinical diagnostic kits for acute and chronic toxoplasmosis [93]. In addition, as a synthetic multiepitope antigen, the recombinant forms of several proteins, including SAG1, ROP2, GRA1, GRA4 and MIC3, have been also considered useful to design a potential ELISA test with specificity and sensitivity of 88.6% and 79.1%, respectively [94].

### 3.5. Triose-Phosphate Isomerase (TPI)

TPI is a glycolytic enzyme in *T. gondii* [95] that provokes common significant lymphoproliferative as well as Th1-biased cytokine responses in both human and golden hamsters infected by other parasites, such as *Leishmania* [96]. There is no information regarding TPI as a vaccine target in toxoplasmosis. However, the present data regarding the activation of the immune system and immunomodulatory properties by this stage-specific protein and the recent suggestions for TPI as a vaccine target in helminth and *Leishmania* parasites might further reinforce the selection of this protein as a promising vaccine candidate against toxoplasmosis [96,97,98,99].

### 3.6. Protein Disulfide Isomerase (PDI)

This is a protein linked to early steps of invasion. It was shown that mice immunized with 30 μg rTgPDI induced high levels of specific antibodies against this protein and protective immune responses (a strong lymphoproliferative response and high levels of IFN-γ, IgG2a, IL-2, and IL-4 were produced) [72].

### 3.7. MICs

Much pathogenic and immunogenic evidence revealed that MIC1, MIC3, MIC4 and MIC6 played a major function in parasite pathogenicity, while MIC3, MIC4, MIC5, MIC6, MIC8 and MIC13 were described as high immunogenic proteins [23].

MIC1-matrix antigen 1 (MAG1) recombinant chimeric antigen can be effectively applied (sensitivity: 90.8%) instead of the *Toxoplasma* lysate antigen (sensitivity: 91.8%) for the serodiagnosis of human toxoplasmosis, exhibiting better results than a mixture of antigens. Additionally, the use of the MIC1-MAG1 protein proposed a promising strategy to identify acute and chronic phases of toxoplasmosis [100].

MIC2 protein complex is a major virulence determinant for *Toxoplasma* infection. It seemed that the transmembrane adhesion MIC2 cooperated with its partner protein M2AP, participating in a major invasion pathway. *MIC2* gene knockout and the decreased expression led to the mistrafficking of M2AP and consequently the loss of helical gliding motility, defective host-cell attachment and invasion, and finally the inability to support lethal infection in a murine model of acute toxoplasmosis. MIC2-deficient parasites acted as an effective live-attenuated vaccine for experimental toxoplasmosis. Furthermore, increased survival rates, a lower parasite burden, decreased inflammatory immune responses and the induction of long-lasting immunity had been observed [101].

The upregulation of MIC3 has been deciphered in pathogenic strains of *Toxoplasma* parasites compared to the less virulent strains [26]. MIC3 was characterized as a protein with a high potential for macrophage M1 polarization and tumor necrosis factor alpha (TNF-α) production. The high expression level of TNF-α in patients with cerebral or ocular toxoplasmosis further confirmed the role of tachyzoites secretions in the induction of TNF-α production [102]. A vaccine strategy based on the prediction of specific epitopes (B cell and T cell) from three *T. gondii* antigens (MRS protein: MIC3, ROP8, and SAG1) has been recently developed in BALB/c mice. Mice immunized with MRS induced stronger humoral and Th1 cell-mediated immune responses in comparison with control mice. Those results proposed that MRS, as a multi-epitope protein vaccination strategy, could be effective against toxoplasmosis infection [103]. Additional results indicated that the application of MIC3 encoding DNA and IL-12 conjugate—a multigene vaccine—might lead to an increase in the Th1 immune responses (increasing the level of IFN-γ) [104].

*MIC1-3* gene knockout induced a strong humoral and cellular Th1 response and induced highly significant protection against chronic infection (>96% reduction in cysts in brain tissue) and congenital toxoplasmosis (fewer infected fetuses in vaccinated groups compared with non-vaccinated (4.6% vs. 33.3%)) [105]. Moreover, it has been indicated that DCs and macrophages are induced by rMIC1 and rMIC4 (through TLR2 and TLR4) driven to the increase in proinflammatory cytokines [106]. Although, in silico data also confirm multiple interesting B- and T-cells epitopes for MIC4 protein [107], more experimental data are needed to corroborate it as a possible vaccine candidate against toxoplasmosis.

Additionally, DNA vaccines encoding *Toxoplasma MIC5* and *MIC16* genes induced effective immunity, including enhanced levels of IgG, IFN-γ, IL-2, IL-12p70, and IL-12p40 and CD4^+^ and CD8^+^ T cells against toxoplasmosis. Moreover, vaccination with such a cocktail vaccine prolonged the mice survival time and decreased brain cysts compared with non-vaccinated groups [108]. The more effective results obtained from the MIC5/MIC16 cocktail vaccine compared to the vaccines containing a single gene of these MICs might further render the use of such approach in MICs-based vaccination against toxoplasmosis.

### 3.8. ROPs

ROP proteins are rhoptry paralogs showing polymorphisms. They are also related with the virulence and the pathogenicity of the different *T. gondii* strains [38]. Many of these proteins are involved in relevant strain specific host immunomodulatory functions, such as in the NFκB-IFN-γ axis or in the antigen presentation by MHC-I for a balanced host immune response required to achieve infection and to reduce CD8^+^ T cell recognition. These data, in addition to the fact that ROP antigens have long antigenic fragments and regions, support their selection as one of the strongest candidates as vaccine antigens [109,110,111]. Several ROP proteins (ROP2, 5, 9, 16, 17, 18, 22, 35) have been employed in vaccine strategies, mainly in DNA or protein vaccines against toxoplasmosis [23,112,113,114]. It has been revealed that the use of ROP1 protein induced high IFN-γ levels but low IL-4 levels in the immunized BALB/c mice [115]. Similar results were observed immunizing with ROP22 protein, however an increase in the survival time of challenged individuals was also reported [113]. A multi-antigenic ROP1 and GRA7 DNA vaccine adjuvanted with IL-12 was able to increase survival (50%) and decrease cyst burden (89%) in the brain of vaccinated mice [116]. Interestingly, ROP4 immunization reduced brain cyst numbers approximately 46% in the rROP4-vaccinated mice [117].

Among other ROPs of the parasite, ROP8, an important protein in *Toxoplasma* proteome, is associated with the *Toxoplasma*-PV, with an unknown function that can be expressed during the early stages of *T. gondii* infection [118,119]. Recently, the co-delivery of a novel multi-epitope plasmid (pc) ROP8 DNA vaccine with a pc encoding IL-12 (pcIL-12) (as a genetic adjuvant) has been evaluated to assess the immune responses in BALB/c mice against acute toxoplasmosis [120]. The results showed the increased level of anti-*Toxoplasma* antibodies (IgG total and IgG2a), Th1-type cellular immune responses (IFN-γ and IL-4), and also a prolonged survival time in immunized mice. Furthermore, vaccination with an ROP21 DNA vaccine also produced high levels of IgG (IgG total, IgG1 and IgG2a) and increased the production of IFN-γ, but the expression of other cytokines (IL2, 4, 10) was not altered [121].

Moreover, cocktailed DNA immunization with ROP5 and ROP18 in combination with adjuvant IL-33 further increased immune responses compared with a single DNA immunization with ROP5 or ROP18. This cocktailed DNA vaccine increased *Toxoplasma*-specific IgG2a titers, Th1 responses correlated with the production of IFN-γ, IL-2, IL-12, and cell-mediated activity with higher frequencies of CD8^+^ and CD4^+^ T cells [122]. ROP18 and MIC6 have also previously been suggested as possible vaccine targets. This vaccine efficiently induced high levels of total IgG, CD4^+^ and CD8^+^ T lymphocytes, and antigen-specific lymphocyte proliferation, and dramatically decreased the parasite cyst burden in vaccinated mice [17]. In addition, ROP18 encapsulated in poly(D,L-lactide-co-glycolide) (PLG) was able to efficiently induce Th1-biased immune responses [123].

### 3.9. Heat Shock Proteins (HSP20 and HSP70)

TgHSP20 is a pellicle-associated functional chaperone localized to the inner membrane complex and to the plasma membrane of the parasite. The incubation of *T. gondii* tachyzoites with an anti-TgHSP20 serum decreased parasite invasion at rates of 57.23% and also reduced parasite gliding by 48.7%, supporting the function of HSP20 in parasite invasion and gliding motility [124]. Such results suggested HSP20 as a possible candidate to design an attenuated vaccine against toxoplasmosis. In addition, the induction of DCs activation and successive early Th1 polarization at draining lymph nodes of C57BL/6 mice by the TgHSP70 protein highlights the immune effects (modulation of the host immune responses) of this protein against toxoplasmosis [125]. TgHSP70 vaccination reduced the inflammation in the brain of infected mice and in parallel anti-rTgHSP70 immune complexes in the serum. Moreover, the induction of inducible nitric oxide synthase (iNOS) expression and the decrease in brain infection were observed in vaccinated mice. It seemed that iNOS expression and consequently nitric oxide (NO) production in the brain was a protective mechanism induced by TgHSP70 immunization [125,126,127].

### 3.10. Toxofilin, Coronin and Peroxiredoxin (Prx)

These proteins were identified in the proteome of the STAgs of the parasite. Toxofilin DNA vaccine combined with the individual adjuvants, aluminum salt (alum) or monophosphoryl lipid A (MPLA), or a mixture of alum-MPLA adjuvant were able to enhance antibody responses against toxoplasmosis. Toxofilin DNA vaccination altered the Th2 immune response to a Th1 response and induced the strongest humoral and Th1 responses. The enhanced survival time and a lower number of cysts were also observed in vaccinated groups [128].

Coronin 1 seemed to be an important regulator of naive T cell homeostasis [129]. Although, a possible role against the host immune defense had been also proposed for coronin in *Toxoplasma* parasites [46], the regulatory function of this actin binding protein concerning the host’s immune cells remains unclear but promising.

Recently, rTgPrx has been applied in dot-immunogold-silver staining (Dot-IGSS) method with a sensitivity of 97.5% and a specificity of 100% to detect *Toxoplasma*-IgG antibodies in infected sera [130]. In addition, the immune-stimulating activity of TgPrx1 included the production of IL-12p40 and IL-6, but not of IL-10, the activation of NF-κB and the induction of specific antibodies (IgG1 and IgG2c) and antigen-specific humoral and cellular immunity [131]. Such information suggested the function of *Toxoplasma* derived redox enzymes, such as Prx, as important immune modulators and probable vaccine and diagnostic candidates for toxoplasmosis [64].

### 3.11. Apical Membrane Antigens (AMAs)

AMA1, only expressed in the *Toxoplasma*-tachyzoite stage, has an immunogenicity and high pathogenicity compared to other AMAs [23]. Three tetravalent chimeric proteins containing different portions of the parasite’s AMA1 antigen-AMA1^domain I^-SAG2-GRA1-ROP1_L_ (A^N^SGR), AMA1^domains II^, III-SAG2-GRA1-ROP1_L_ (A^C^SGR) and AMA1^full protein^-SAG2-GRA1-ROP1_L_ (A^F^SGR)- were evaluated for their immunogenic and immunoprotective potentialities [132]. All evaluated proteins were immunogenic and triggered specific humoral and cellular immune responses in vaccinated mice. However, the intensity of the produced immune protection depended on the fragment of the AMA1 antigen incorporated into the chimeric antigen’s structure. It has been identified that full length AMA1 can trigger further potent immunity in mice, resulting in significantly increased survival and partial protection against *Toxoplasma* cyst formation [132]. Furthermore, the full-length AMA1 and two different fragments (AMA1N and AMA1C) have been tested for the detection of IgG and IgM anti-*Toxoplasma* antibodies in human and mouse immune sera in ELISA assays [133]. The results demonstrated that the full-length AMA1 recombinant antigen (corresponding to amino acid residues 67–569 of the native AMA1 antigen) is a better biomarker (reacting with specific anti-*Toxoplasma* IgG (sensitivity: 99.4%) and IgM (sensitivity: 80.0%) antibodies) for the diagnosis of toxoplasmosis in comparison with the C- or N-terminal fragments of the antigen.

### 3.12. Protein Phosphatase 2C (PP2C) and Altered Thrombospondin Repeat Domain (SPATR)

*T. gondii* can deliver PP2C into the host cell and direct it to the host cell nucleus [134]. It has been shown that the immunization with the rPP2C significantly induced specific IgG antibodies and cytokines and also enhanced the survival rate of immunized mice compared with that of the control groups making this a potential vaccine candidate against acute toxoplasmosis [67]. Furthermore, a SPATR-based vaccine generated humoral and mixed Th1/Th2 type cellular immune responses inducing lymphocyte proliferation and cytokine (IFN-γ, IL-2, IL-4 and IL-10) secretion, showing that SPATR may be a promising vaccine candidate against toxoplasmosis [135].

### 3.13. Myc-Regulating Protein 1 (MYR1)

*Toxoplasma* parasites deficient in MYR1 induced a weak pathogenicity in mouse infection models, suggesting that MYR1 decisively influences parasite delivery of effector proteins to the infected host cells [136]. Thus, rMYR1 protein has been suggested as a potential DNA vaccine candidate that activated Th1 and Th2 T-cell response (increasing significant levels of Th1 and mixed Th1/Th2 cytokines) at two and six weeks after immunization, respectively [70].

### 3.14. Embryogenesis-Related Protein (ERP)

ERP belongs to a group of four molecules called late embryogenesis abundant domain-containing proteins (LEAs). ERP as a protein specifically expressed in sporozoites of *T. gondii* might be used in the differentiation of tissue cyst-induced toxoplasmosis from oocyst-induced toxoplasmosis in mice, pigs, and humans [50] and consequently allowing the accurate identification of the source of infection. The seroepidemiological aspects of ERP protein was recently described [137]. For instance, the use of anti-TgERP salivary IgA for the estimation of the prevalence of toxoplasmosis in endemic areas (in individuals 15–21 years old) has shown satisfactory results (a specificity of 93.33% and sensitivity of 93.94%) [138].

In addition to the abovementioned proteins, proteomic analyses have also recognized several proteases, including cathepsins, leucyl and aspartyl aminopeptidases, prolyl endopeptidase and serine protease in the *Toxoplasma*-tachyzoites ESAs [139]. Some *Toxoplasma*’s proteases, such as cathepsin C1 and aspartic protease 3, have been described as enzymes with immune-protector roles in toxoplasmosis [140,141]. In addition, serine proteases have been also underlined as potential vaccine candidates in other parasitic diseases [142,143]. ROMs are a class of serine proteases that play major functions in parasite (such as *Toxoplasma*) invasion and in mitochondrial fusion and growth factor signaling, allowing the parasite to facilitate the entrance into the host cell [144]. According to the roles of proteases, especially in parasites ESAs, such enzymes could be further considered as vaccine targets.

## 4. Perspectives with Proteins Expressed in other Structures of *T. Gondii*

Some other proteins have been identified in the proteome of important structures of the *Toxoplasma* parasite (Figure 3) [145,146,147,148]. Besides their possible therapeutic properties [145,146,149,150], such novel proteins might be further evaluated as diagnostic and vaccine targets against toxoplasmosis in the future. Since all these structures and compartments may be very dynamic at the molecular level, thus adding complexity for confidently assigning proteins to a specific subcellular compartment, an expansion of known organelle proteomes has been recently conducted applying modern spatial proteomic methods [35]. The use of such advanced tools can further identify novel proteins relevant to the parasite organelles with possible critical and vital functions in the biology and pathogenesis of *Toxoplasma* parasites. The subpellicular cytoskeleton is a vital structural part of the *Toxoplasma* parasite involved in the motility, invasion, and maintenance of the shape of different forms of the parasite [151,152,153].

The investigation related to the proteome of the *Toxoplasma* subcellular niches is essential to understand their functions which might increase our knowledge regarding parasite pathogenicity and may also lead to introducing novel diagnostic and vaccine targets against toxoplasmosis. TgGRA8_I_, expressed in the *Toxoplasma* subpellicular cytoskeleton proteome, plays important role in the formation of the PV and participates in the organization of the *Toxoplasma* subpellicular cytoskeleton and motility of this parasite [148]. TgGRA8 has been recently demonstrated as a possible serological biomarker for detecting specific *Toxoplasma*-IgG in goat sera. The sensitivity and specificity of the LAT for the recombinant form of this protein were 71.1% and 96.0%, respectively [154]. Moreover, elongation factor 1-alpha (EF-1α) has been identified in the proteome of *Toxoplasma*-subpellicular cytoskeleton, playing an important function in mediating host cell invasion by the parasite [148]. Relevant results on the evaluation of vaccine efficacy of EF-1α indicated significantly increased survival time (14.53 ± 1.72 days) of infected mice after challenge infection with the virulent *T. gondii* RH strain [155].

The dynamic adhesion, invasion, and even replication properties of *Toxoplasma* are based on machinery located in the pellicle. A group of glycosylphosphatidylinositol (GPI)-linked proteins (SRSs) were identified as important proteins in *Toxoplasma*-pellicle proteome [156]. *Toxoplasma* SRS13 and SRS29A have shown strong immunogenicity but have not been evaluated in the development of a vaccine model or diagnostic assay yet [157]. Furthermore, the use of rSRS3 protein in an ELISA system represented a sensitivity and specificity 84.12% and 92%, respectively [158].

## 5. Immune Response-Based Candidates for Disease Management

Proteins selected as diagnostic, or vaccination candidates are required to comply with some aspects that ensure their intended activity. As the mechanisms explaining how diagnosis and vaccination in parasites work seem different, then the ideal characteristics required for a target may also exhibit variations. Hence, by applying this perspective, we suggest a categorization of the proteins based on the role they would fulfill better. According to the literature revised in this manuscript, most of the methods that exploit the proteome of *Toxoplasma* for diagnosis require from the recognition of an antigenic determinant by an antibody [123,159]. Therefore, it would be suggested to select immunodominant proteins in which the antibody production predominates rather than cellular response. On the other hand, the response needed for an effective vaccination is more complex and an optimal target should develop a more cellular related immune response. These vaccination candidates should generate a Th1 immune response with the production of high levels of IFN-γ and IL-12 which induce the effectors that directly neutralize *Toxoplasma* or suppress its growth [160]. Table 2 summarizes potential candidates for both diagnosis and vaccination, highlighting their main features for the more suitable purpose. As the lack of standardization of a vaccination and protection assays hinders the cross-study comparison of results, it would be inaccurate to rank the best proteins. However, it would be important to remark on those candidates that stand out from the rest, especially when the immune response is well characterized. Both GRA7 in synergy with profilins [82] and GRA9 [81] elicited a strong Th1 related response and the production of proinflammatory cytokines, resulting in higher rates of protection.

**Table 2 animals-12-01098-t002:** Potential stage-specific vaccine and diagnostic candidates in toxoplasmosis.

Proteins	Vaccine/Diagnostics Utility	Vaccine/Diagnostics Efficacy	References
GRA4	Edible vaccine	Eliciting both mucosal (the production of specific IgA, and IFN-γ, IL-4 and IL-10 secretion by mesenteric lymph node cells) and systemic (in terms of GRA4-specific serum antibodies and secretion of IFN-γ, IL-4 and IL-10 by splenocytes) immune responses	[77]
GRA5	Diagnostic tool	Specificity: 93%, sensitivity: 96%	[78]
GRA6	DNA vaccine ADJ with LMS	High levels of anti-GRA6 IgG and splenocyte proliferation	[80]
GRA7	Live-attenuated vaccine ADJ with profilin	Enhancing expression of CD80 and CD86 in BMDCs and secretion of IL-6, IL-10 and IL-12 Eliciting a Th1-biased immunity through the induction of lymphocyte proliferation, activation of CD4^+^ T cells and increased IFN-γ production	[82]
GRA9	Live-attenuated vaccine	Inducing high levels of IFN-γ, IL-12, and IgG1/IgG2a levels (100% protection)	[81]
GRA14	DNA and Recombinant vaccine ADJ with CaPNs	Increasing antibody titers (increased levels of total IgG and IgG2a) and concentration of IFN-γ (a Th1 type response)	[86]
GRA1 + GRA7 + ROP2	DNA vaccine	Inducing Th1 response (a high ratio of specific IG2a to IgG1), increasing survival rate from 50% to at least 90%, decreasing the number of brain cysts	[76]
GRA1 + GRA4 + GRA6 + GRA7	DNA vaccine formulated into liposomes	GRA7: Inducing anti-GRA7 IgG2 and IFN-γ (Th1-like immune response), GRA1, GRA4 and GRA6: stimulating a IgG1 type antibody response with a limited IFN-γ response	[79]
SAG1	Recombinant SAG1 vaccine (encapsulated in PLGA nanosphere)	Eliciting elevated humoral responses of specific IgA and IgG2a	[89]
	rSAG1 (diagnostic tool)	Sensitivity and specificity of 98.5% and 100%, respectively	[92]
SAG1 + apicoplast ribosomal proteins + human TLR-4	Multi-epitope vaccine	Inducing humoral (T- and B-cell mediated responses) and cellular (high levels of IFN-γ) immune responses	[91]
SAG1 + GRA7 + ROP1	Diagnostic tool	Sensitivity and specificity (undetermined)	[93]
SAG1 + ROP2 + GRA1 + GRA4+ MIC3	A synthetic multiepitope antigen (diagnostic tool)	Specificity: 88.6% and sensitivity 79.1%	[94]
ROP1	DNA and Recombinant vaccine	Inducing high IFN-γ level but low IL-4 level in the immunized mice	[115]
ROP4	Recombinant vaccine	Inducing specific production of IFN-γ as well as IL-2, the Th1-type cytokines, reducing brain cysts number approximately 46% in the rROP4-vaccinated mice)	[117]
ROP5 + ROP18	Cocktail DNA vaccine	High specific IgG2a titers, Th1 responses correlated with the production of IFN-γ, IL-2, IL-12, and cell-mediated activity with higher frequencies of CD8^+^ and CD4^+^ T cells	[122]
ROP8	DNA vaccine ADJ with IL-12	Increasing the level of anti-*Toxoplasma* antibodies (IgG total and IgG2a), Th1-type cellular immune responses (IFN-γ and IL-4), lymphocyte proliferation, and also prolonged survival time in the immunized mice	[120]
	Diagnostic tool (using Western blotting technique)	In early acute (sensitivity 90%), acute (sensitivity 92%), and chronic toxoplasmosis (sensitivity 82%) (specificity 94% for all stages)	[161]
ROP1 + GRA7	Multi-antigenic DNA vaccine ADJ with IL-12	Increasing serum IgG2a titers, production of IFN-γ, IL-10, and TNF-α (increasing survival (50%) and decreasing cyst burdens (89%) in the brain of vaccinated mice)	[116]
ROP18, MIC6, in combination with PF, ROP16, and CDPK3	Cocktail DNA vaccine	Eliciting a mixed Th1/Th2 response, with a slightly elevated IgG2a to IgG1 ratio, the enhanced production of proinflammatory cytokines IL-2, IL-12 and IFN-γ, reduction in the parasite cyst burden (80.22%)	[17]
ROP18 encapsulated in PLG	Recombinant vaccine	Inducing Th1-biased immune responses, with enhanced specific antibodies and T cells, high levels of INF-γ and IL-2, and strong lymphocyte proliferative responses	[123]
MIC1-MAG1	Diagnostic tool	Sensitivity: 90.8%, specificity: 100%	[100]
MIC2	Live-attenuated vaccine (MIC2-deficient)	Increasing survival of vaccinated mice correlated with lower parasite burden in infected tissues, decreasing inflammatory immune response, and induction of long-term protective immunity	[101]
MIC3	DNA vaccine ADJ with IL12	Increasing the level of IFN-γ	[104]
MIC1-3	Live-attenuated vaccine	Inducing humoral and cellular Th1 response, >96% reduction in cysts in brain tissue	[105]
MIC5/MIC16	Cocktail DNA vaccine	Enhanced levels of IgG, IFN-γ, IL-2, IL-12p70, and IL-12p40 and CD4^+^ and CD8^+^ T cells, and prolonged mice survival time and decreased brain cysts (48.06%)	[108]
AMA-1	Diagnostic tool (ELISA)	Reacting with specific anti-*Toxoplasma* IgG (sensitivity: 99.4%) and IgM (sensitivity: 80.0%)	[133]
	Recombinant epitope vaccine	Inducing Th1/Th2 cytokines, the production of IgG1/IgG2a, increasing survival and partial protection against parasite-cyst formation	[132]
ADF	Recombinant vaccine	The increased levels of IgG, IL-2 and IFN-γ, increasing survival rate (36.36%) and decreasing tachyzoite load in the liver (67.77%) and brain (51.01%)	[74]
NTPase-II	RNA vaccine	Inducing IgG and IFN-γ, prolonged survival time, reducing parasite load in the brain (46.4% and 62.1% in acute and chronic infections, respectively)	[75]
HSP70	Recombinant vaccine ADJ with alum	Reducing inflammation in the brain and anti-rHSP70 immune complexes in serum, inducing iNOS expression and decreasing brain parasitism	[127]
Toxofilin	DNA vaccine ADJ with alum-MPLA	Changing Th2 to a Th1 response and provoking the humoral and Th1 responses, inducing survival time and decreasing cyst ratio	[128]
SPATR	DNA vaccine	Activating humoral and mixed Th1/Th2 cellular responses (inducing IFN-γ, IL-2, IL-4, and IL-10)	[135]
PP2C	DNA vaccine	The increased levels of IgG2a (a predominantly Th1 immune response) and cytokines (IFN-γ)	[67]
PDI	Recombinant vaccine	Inducing higher levels of IFN-γ, IgG2a, IL-2, and IL-4	[72]
MYR1	DNA vaccine	Increasing significant levels of Th1 and mixed Th1/Th2 cytokines	[70]
ERP	Diagnostic tool (ELISA)	Specificity: 93.33%, sensitivity: 93.94%	[138]
Prx	Diagnostic tool (Dot-IGSS)	Sensitivity 97.5% and specificity 100%	[130]
	Recombinant vaccine	Triggering IL-12p40 and IL-6, the activation of NF-κB, eliciting specific antibodies (IgG1 and IgG2c)	[131]

Levamisole (LMS), adjuvant/adjuvanted (ADJ), bone marrow-derived DCs (BMDCs).

Some candidates have been tested both as vaccination and diagnostic candidates. For example, SAG1 [89], ROP8 [120] and AMA-1. They produced an important humoral response with an increased production of IgGs when injected as vaccination candidates. As the response was predominantly humoral, they were found to be very suitable targets for diagnosis by ELISA [92], Dot-IGSS [130] or Western blot [161]. Hence, our insistence on the difference between a humoral or cellular directed response when choosing a *Toxoplasma* protein as a target. Furthermore, Table 2 also shows the type of vaccine that has been used in the reported evaluation of vaccine efficacy. It is well known that several candidates have already been evaluated using more than one vaccination technology. We need to keep in mind that it is not the topic of this review to discuss the vaccine technology employed on each of those candidates. On the other hand, there are authors that have recently addressed that issue, such as Mamaghani et al., 2022 [111].

## 6. Conclusions and Future Directions

Recent technological improvements for the study of proteome alterations during *T. gondii* life stage conversions throughout the sexual cycle have led to further answers to biological questions related to *Toxoplasma*-life cycle stages, and will probably open new insights towards effective vaccines [162,163]. Accordingly, the present study reviewed the vaccine and diagnostic properties of functionality important proteins expressed in the life cycle-specific stages of *Toxoplasma* parasites identified, applying proteomic approaches. The proteomics applied for the identification of key parasitic structures also provide valuable sources of functional proteins in these parasites. All these targets open new avenues and may shed some light on biological features of *Toxoplasma*, such as survival, pathogenicity, metabolic pathways, parasite-host interactions, and its life cycle. As one final aim, this information may help the reader to understand the complexity of these parasites and the potential of many proteins to initially rise good expectations as diagnostics or vaccine candidates to control toxoplasmosis.

It seems that, according to the heterogeneity of host immune responses against *Toxoplasma* infection and the possible challenges for selecting appropriate diagnostic markers, the combination of immunogens (synthetic multiepitope antigen) may be useful for the design of diagnostic tests in human toxoplasmosis [94,164]. In addition, recent advances in our knowledge of parasite genetics and gene manipulation, key antigenic epitopes, strain variation, delivery systems and induction of immune responses are considered participating insights for the development of new vaccines which may be more efficient against toxoplasmosis [165].

Traditionally, *T. gondii* vaccination and diagnostic candidates have been selected by experimentally testing the immunity produced by proteins isolated directly from the pathogens using costly and time-consuming techniques. In 2001, the vaccine against serogroup B meningococcus was developed by using genome information and the “reverse vaccinology” was born [166]. This strategy used computational methods in silico to predict the suitability of a gene, protein, or epitope as vaccine candidates, allowing for high-throughput screening of “omics” data. The recently suggested genome-wide comparative datasets analyses integrating Open Reading Frame (ORF)-mediated translational regulation may reveal genomic variants important for stage conversion and thus novel parasite-specific, essential proteins not previously detected by proteomics because of the low levels for proteins coded by repressive upstream ORFs containing mRNAs. Some of these may have the likely potential to be considered as diagnostics and even vaccine candidates [167,168,169,170].

Advanced in silico models are being developed that estimate several characteristics, such as MHC bind capacity [171], T-cell receptor recognition [172], immunogenicity [173], subcellular location [174,175], etc. The implementation of these tools in machine learning models that unify estimates for several features is paramount to develop integrated computational pipelines to profile and characterize classical and new vaccination targets for *T. gondii*, similar to the approach recently applied in cancer derived neoantigens [176]. Regarding the immense number of *T. gondii* proteins reviewed in this manuscript, we suggest a further in-depth analysis using ad hoc machine learning models that integrate parasite data.

Most of the research revised in this manuscript selects one or a small number of proteins that provide several degrees of partial protection. For this reason, some scientists believed that vaccines for complex pathogens, such as *T. gondii*, will not produce total protection using a single candidate. In silico models with the use of machine learning could help with the task of developing a more effective vaccine by characterizing and predicting the most immunogenic epitopes of proteins [177] and working towards a multiepitope vaccine. Vaccination with epitopes in *T. gondii* has recently been addressed [178] but the topic is fast-evolving and the prediction of epitopes and its ability to predict its binding strength to MHC improves continuously. Therefore, we encourage the revisiting of all the proteins addressed in this manuscript using present-day techniques with machine learning models [179] to predict the most immunodominant epitopes and assay them in a multiantigen vaccine, seeking a fully protective multi-antigen, multi-stage vaccine.

The inclusion of multi-epitopes seems to enhance the specificity of antigenic and antibody responses and along with in silico approaches may facilitate important advances within a “one health” perspective [162,178]. To this end, the progress in proteomics needs to assess reliable protein characterizations and fully using the power of all the modern proteomic setups, as well as to explore combinatorial and new developments. In addition to the proteomic tools, other new tools aiming to identify protein composition of the different *T. gondii* stages, including oocyst and cyst walls and stage conversions, such as interactome constructions using proteins identified via BioID or RNA single cell sequencing [180], could lead to a better understanding of the parasite biology and introduce possible novel vaccine candidates for multi-antigenic, more effective, vaccines [11].

## Figures and Tables

**Figure 1 animals-12-01098-f001:**
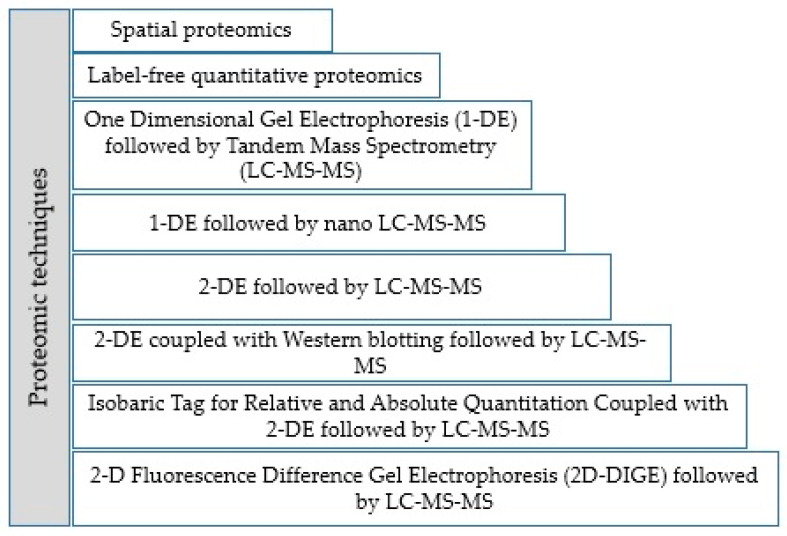
Proteomic techniques mainly used for the identification of *T. gondii* proteome.

**Figure 2 animals-12-01098-f002:**
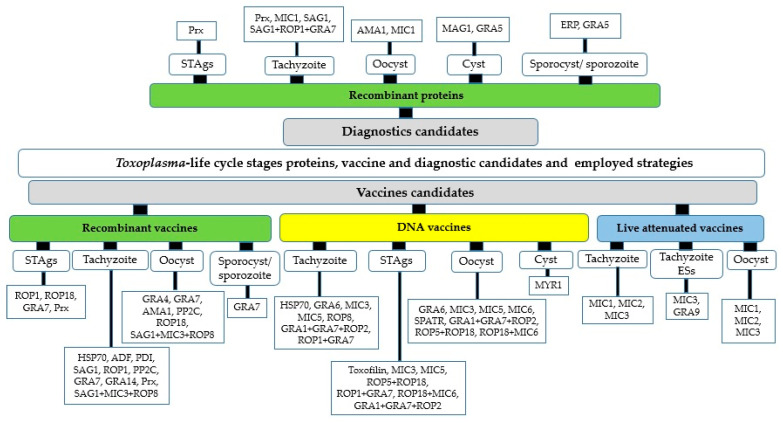
*Toxoplasma* vaccine and diagnostic candidates inferred from proteomics data.

**Figure 3 animals-12-01098-f003:**
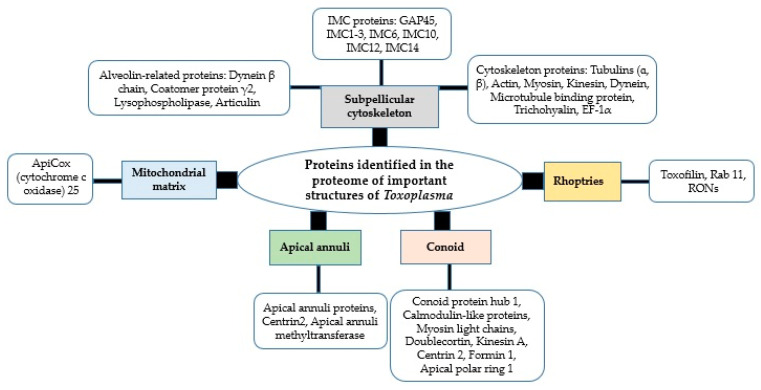
Proteins identified in the proteome of important structures of the *Toxoplasma* parasite.

**Table 1 animals-12-01098-t001:** The biological functions of proteins expressed in life cycle-specific stages of *Toxoplasma* parasites employed as vaccine candidates.

Proteins	Location	Biological Functions	References
ADF	A related actin-binding protein (cytoskeleton)	Remodeling the actin cytoskeleton (increasing the actin filaments turnover) and parasite host cells invasion	[54,55]
NTPases	Dense granules	Processing of nucleotides for purine salvage by the parasite, parasite replication and virulence	[56,57]
GRAs	Dense granules	The alteration of PV and the PV membrane in parasite (maintenance of intracellular parasitism in host cells)	[58]
SAG1	Parasite surface antigen	Recognition, adhesion and invasion of host cells	[59]
TPI	Carbohydrate metabolism cycle	A virulence factor with important roles during pathogenesis via glucose levels modulation	[60]
ROPs and RONs	Rhoptry	Participates in the moving junction formation during parasite invasion	[61]
Toxofilin	A secretory protein from rhoptries	Binds to the parasite and mammalian actin and plays role in the host cell invasion	[62]
Prx	A redox enzyme probably in parasite nucleus	Phagocytosis, transcriptional regulation, receptor signaling, and protein phosphorylation, maintenance of parasite oxidative balance	[63,64]
AMA1	Microneme	Host cell recognition and attachment	[65]
SPATR	Microneme	Parasite virulence and host cell recognition	[66]
PP2C	Rhoptry	Targeting the host nucleus and plays a role in parasite invasion	[67]
MIC3	Microneme	A predominant role in the early phase of the invasion process	[68]
MYR1	PV membrane	Exporting parasitic proteins, parasite pathogenesis	[69,70]
ERP		Related to the resistance of parasite (oocyst) against environmental stresses	[51]
HSP20	IMC (parasite plasma membrane)	Protect and/or modulate membrane properties of the IMC	[71]
HSP70		A potential immunoregulator (B cell mitogen and inducing DC maturation)	[38]
PDI	Surface of tachyzoites	Host cell interactions	[72]
MAG1	A protein in PV matrix, in tachyzoite vacuoles and the cyst wall and matrix in bradyzoite vacuoles	As an immunomodulatory molecule (suppressing inflammasome activation)	[73]

Parasitophorous vacuole (PV), inner membrane complex (IMC), dendritic cell (DC).

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
