# Peer review of "Mining the Proteome of Toxoplasma Parasites Seeking Vaccine and Diagnostic Candidates"

_animals, 2022, doi:10.3390/ani12091098_

Round 1

Reviewer 1 Report

The current study might help to understand the complexity of these parasites and provide a possible source of strate-gies and biomolecules that can be further evaluated in the pathobiology of Toxoplasma parasites and for diagnostics and vaccine trials against this disease.

However, The structure of the article is a little chaotic and disorganized, and it only lists the published vaccine candidate antigens, without further exploring the potential highlights. This article can review the research situation of Toxoplasma gondii vaccine from different stages of vaccine development (Attenuated vaccine, whole worm protein or secretory protein vaccine, genetic engineering vaccine, subunit vaccine, nano vaccine),so that readers can better understand it.

The literature consulted in this review article is not comprehensive enough. For example, the immune functions of Toxoplasma gondii rop21, rop22 and rop33 have been reported, but they are not found in the manuscript. 

Author Response

Thank you.

Reviewer 2 Report

The authors have summarized possible antigen candidates for vaccines and diagnostic antigens for Toxoplasma. This information may be useful to readers in the relevant research areas. However, some revisions need to be made before acceptance.

Major;

  1.  The authors should explain each apical organelle using a figure BEFORE explanation of Tg proteins.
  2.  All 'Toxoplasma', 'T. gondii', 'in silico' must be written by italic.

Minor;

  1.  Improve image resolution of Figure 2 and 3.
  2.  'MIC3' on Line 242 is bigger than other letters.
  3.  'T. gondiii' on Line 260 must be 'T. gondii'
  4.  No need to be italic on Line 345 and Line 399.

Author Response

Thank you.

Round 2

Reviewer 1 Report

The manuscript can be accepted with language modification.

Reviewer 2 Report

The authors have satisfyingly answered all my comments.